# Immune Modulation by Inhibitors of the HO System

**DOI:** 10.3390/ijms22010294

**Published:** 2020-12-30

**Authors:** Ayleen Fernández-Fierro, Samanta C. Funes, Mariana Rios, Camila Covián, Jorge González, Alexis M. Kalergis

**Affiliations:** 1Millenium Institute on Immunology and Immunotherapy, Departamento de Genética Molecular y Microbiología, Facultad de Ciencias Biológicas, Pontificia Universidad Católica de Chile, 8331150 Santiago, Chile; alfernandez1@uc.cl (A.F.-F.); mrios@bio.puc.cl (M.R.); c.covianm@uc.cl (C.C.); jigonzalez10@uc.cl (J.G.); 2Instituto Multidisciplinario de Investigaciones Biológicas-San Luis, Consejo Nacional de Investigaciones Científicas y Técnicas—Universidad Nacional de San Luis, 5700 San Luis, Argentina; samanta.funes@gmail.com; 3Departamento de Endocrinología, Facultad de Medicina, Pontificia Universidad Católica de Chile, 8331150 Santiago, Chile

**Keywords:** heme oxygenase-1, heme oxygenase-2, inhibitors, immunomodulation, infections, cancer

## Abstract

The heme oxygenase (HO) system involves three isoforms of this enzyme, HO-1, HO-2, and HO-3. The three of them display the same catalytic activity, oxidating the heme group to produce biliverdin, ferrous iron, and carbon monoxide (CO). HO-1 is the isoform most widely studied in proinflammatory diseases because treatments that overexpress this enzyme promote the generation of anti-inflammatory products. However, neonatal jaundice (hyperbilirubinemia) derived from HO overexpression led to the development of inhibitors, such as those based on metaloproto- and meso-porphyrins inhibitors with competitive activity. Further, non-competitive inhibitors have also been identified, such as synthetic and natural imidazole-dioxolane-based, small synthetic molecules, inhibitors of the enzyme regulation pathway, and genetic engineering using iRNA or CRISPR cas9. Despite most of the applications of the HO inhibitors being related to metabolic diseases, the beneficial effects of these molecules in immune-mediated diseases have also emerged. Different medical implications, including cancer, Alzheimer´s disease, and infections, are discussed in this article and as to how the selective inhibition of HO isoforms may contribute to the treatment of these ailments.

## 1. Introduction

Heme oxygenase (HO) is an enzyme that was originally described in 1969 that catalyzes the oxidation of the heme group to form biliverdin, an intermediated step to finally form bilirubin through the NADPH-dependent biliverdin reductase [1]. Later, it was found that HO also generates carbon monoxide (CO) and ferrous iron (Fe^2+^) as reaction products [2,3]. HO and their reaction products have antioxidant, anti-inflammatory, and signaling activity [4,5]. There are three isoforms of HO in mammals, HO-1, HO-2, and HO-3, and a fourth isoform in plants [6]. Even though all isoforms catalyze the same reaction with similar efficiencies and their C-terminal region acts as an anchor to the membrane, they have some structural differences. Thus, while HO-2 and HO-3 contain two heme regulatory motifs (HRMs) with cysteine residues that bind to the heme group independently of the core, HO-1 lacks these HRMs [3,7].

Studies referring to the distribution of the HO system have shown that HO-1 is expressed ubiquitously, being predominantly found in the liver and spleen [8]. This isoform works as an inducible protein, upregulated by different stimuli, such as heavy metals, growth factors, cytokines, and heme, among others [8,9]. On the other hand, HO-2 is constitutively expressed at high levels in the brain, testes, or endothelial cells [8,9], while HO-3 has been observed only in rat astrocytes, although at lower levels than HO-2 [10]. It is thought that HO-3 is a retrotransposition (pseudogene) of the HO-2 gene [10]. Altered HO-1 expression is reported in many clinical conditions, such as autoimmune diseases [11], porphyria, obesity [12], cancer [8,13], and infection diseases, among others [12]. This article will focus on pathologies caused by the overexpression of HO-1 and how inhibitors can be beneficial due to their immunomodulatory activity.

## 2. Regulation of the HO Activity

Understanding the molecular mechanisms involved in the regulation of the expression and activity of the HO enzyme (Figure 1) is essential to define how the current inhibitors work and to develop novel compounds. Transcripts of HO-1 and HO-2 are encoded by the *HMOX1* and *HMOX2* genes, respectively [14,15]. The natural substrate of HOs is heme, and the oxidation of this compound generates CO, which plays a protective and antioxidant role during physiological and pathological conditions [16]. Fe^2+^, also a product of the enzymatic degradation of the heme group, contributes to the regulation of cellular function, essentially as this metal is required for ferritin and hemoglobin synthesis [17]. Biliverdin, the third product of the HO-catalyzed reaction, is transformed to bilirubin in a reaction catalyzed by the biliverdin reductase (BVR) [8].

Under specific conditions, such as cellular stress, HO-1 can be regulated by the antioxidant response element (ARE) located at the HO-1 gene promoter that binds to the nuclear factor erythroid 2-related factor 2 (Nrf2) and promotes the expression of this gene [18]; at the same time, levels of Nrf2 are directly controlled by NFκB expression, which is overexpressed during inflammatory processes [19]. Similarly, the activator protein 1 (AP-1) transcription factor responds to oxidative stimuli by binding to enhancers flanking the *HMOX1* protomer region and increases HO-1 transcription in both immune and non-immune cells [11]. Signal transducer and activator of transcription 3 (STAT3) is another transcription factor able to promote HO expression [20]. IL-10 and also IL-6 are cytokines known to activate STAT3, which requires the activation of the phosphatidylinositol-3 kinase (PI3K) pathway [20]. The mitogen active protein kinase (MAPK) pathway corresponds to a well-known signaling pathway leading to HO-1 expression, principally in response to hypoxia [21]. Furthermore, the length of a (GT)n dinucleotide repeat in the promoter region of HO-1 exhibits variable transcription capacity, correlating a long length with a poor transcription while a short length is associated with an increased HO-1 transcription [22].

On the other hand, HO-2 activity is substrate dependent [23]. Therefore, as heme substrate availability increases, the oxidative reaction will also increase [23]. In addition, post-translational modifications, such as phosphorylation at Serine 79, can also increase the enzymatic activity of HO-2 [23]. In contrast, inhibition of 25–60% of the enzyme activity results after the binding of nitric oxide (NO) species to cysteine residues (Cys265 and Cys282) located at the C-terminal region of the HRMs of HO-2 [24,25,26]. Interestingly, NO species increase HO-1 mRNA and protein levels without changing the protein amount for HO-2, suggesting a compensatory effect [27,28].

Although HO-1 has been much more studied with regards to an up- or downregulation by pharmacological treatments [12], HO-2 has not been characterized with equivalent depth. However, it would be important to also consider HO-2 as a potential therapeutic target. In the next section, several compounds that inhibit HO activity will be discussed.

## 3. Inhibitors of the HO System

As described above, the HO system catalyzes the degradation of the heme group, generating CO, biliverdin, and Fe^2+^ as products [29]. Several studies have described some of these products as anti-inflammatory agents for many chronic and infectious diseases [10,11,12,30]. However, to understand the function of this enzymatic system, the identification and synthesis of inhibitory molecules were necessary [31,32]. The first generation of HO inhibitors consisted of organic molecules composed of four pyrrole subunits with a central metal ion, including protoporphyrins (PPs) and mesoporphyrins (MPs) [33]. Even though these molecules represent a promissory alternative for treating clinical conditions originated by the overexpression of OH, such as hyperbilirubinemia [34], the occurrence of negative collateral effects has been observed [35]. Among these effects, the oxidation of molecules and even death because of their photosensitive activity has been described in preclinical studies [35,36]. The second-generation HO inhibitors consist mostly of synthetic imidazole-dioxolane molecules derived from Azalanstat, which has been shown to efficiently inhibit HO activity [37,38]. Currently, both generations of inhibitors are undergoing evaluation for the treatment of diseases, such as cancer [39,40], neurodegenerative diseases [41,42,43], and infections [44,45,46,47]. As research evolves, the modulation of HO activity has also been studied by using various approaches, such as the generation of novel small molecules with an allosteric inhibitory capacity [48]. Furthermore, by genetic engineering techniques, the selective knockdown or knockout of a specific isoform of the enzyme has been accomplished [7,49]. An updated list of various HO inhibitors is provided in Table 1.

### 3.1. First Generation of HO Inhibitors

Porphyrins are widely and naturally found in mammals, and the main nucleus of these molecules is a cyclic tetrapyrrole able to coordinate with metal ions [50]. The heme group is a protoporphyrin associated with Fe^2+^ (Figure 2A), which forms the prosthetic group of the hemoglobin protein and is the substrate for the enzymatic activity of HO [50]. Metalloporphyrins are the first generation of HO inhibitors [33]. Based on the porphyrin structure, several synthetic PPs, and MPs (Figure 2A) linked to different metals have been tested for the ability to competitively inhibit heme group degradation [33,38,51]. Studies have shown that Tin- (Sn-), Zn-, and Mn-protoporphyrins can strongly inhibit heme degradation in a declining order [33].

Tin protoporphyrin (SnPP) appears to be one of the most potent inhibitors of HO activity, which is rapidly cleared from plasma and persists in tissues, principally in the kidney and liver, inhibiting the HO system for prolonged periods of time [67]. SnPP has also been shown to be the only PP capable of crossing the blood-brain barrier (BBB) [52]. Nevertheless, SnPP and other metallo-PPs are photosensitizers that can cause oxidation of molecules, such as membrane lipids, energy metabolites, proteins, and nucleic acids [52]. Despite these potential undesired effects for patient treatment, SnPP has been evaluated as an antiviral drug for various viruses, such as HIV-1 [68], hepatitis B and C [69,70], as well as nonenveloped viruses (i.e., poliovirus) [71].

After searching for a non-photoreactive inhibitor, Cr- and Mn- porphyrins showed no phototoxicity in vitro and in vivo, with CrMP being the most potent in vitro HO inhibitor [52]. Finally, this first generation of inhibitors has shown disadvantages, such as a lack of selectivity for a particular HO isoform [23,53]. Furthermore, enzymatic activity measurement is important because some metalloporphyrin can enhance HO-1 mRNA synthesis as a compensatory effect [23]. These inhibitors have also been shown to modulate other enzymes, including cytochrome P450 (CYP450), nitric oxide synthase (NOS), and soluble guanylyl cyclase (sGC) [72].

### 3.2. Second Generation of HO Inhibitors

Novel non-competitive inhibitors were necessary to avoid cross-reaction with other enzymes associated with the porphyrins prosthetic group. Azalanstat, a synthetic imidazole firstly designed as an inhibitor for lanosterol 14α-demethylase (14-DM), a member of the CYP450 superfamily, also displayed an inhibitory effect over HO [37]. As shown in Figure 2B, the structure of azalanstat contains four domains, and while the eastern region interacts with the Fe^2+^ from the heme group, the western region is responsible for the HO-1 and HO-2 selectiveness [38,54].

Multiple imidazole-dioxolane compounds derived from the azalanstat structure were introduced, originating the second generation of HO inhibitors [55]. Many of these compounds showed non-selective inhibition after the dioxolane ring was removed [54]. Since several antifungal agents have similar structures with imidazole-derived HO inhibitors, it was proposed that ketoconazole, terconazole, and sulconazole could inhibit HO activity [56,57]. Results have suggested that all of them are effective inhibitors of the activity of both HO-1 and HO-2 in vitro at therapeutic drug concentrations but with significantly higher inhibitory capacity over HO-1 [51,57].

Several studies altering the azalanstat principal structure led to the synthesis of an isozyme-selective inhibitor [58,59,73]. A pool of highly selective inhibitors with substitutions at position 4 of the dioxolane ring showed significant inhibition of HO-1 activity [73]. Furthermore, clemizole synthesis led to one of the main compounds belonging to the first series of highly selective HO-2 inhibitors [58]. Based on these studies, a new series of 1,2-disubstituted benzimidazoles were made with improved inhibitory activity against HO-2 [59].

Considering that the imidazole nucleus is largely recognized as hepatotoxic, the creation of a database, including HO-1 and HO-2 inhibitors known as HemeOxDB [74], has allowed elucidation of alternatives for replacements [74]. A series of natural compounds based on a statistical/computational approach were identified as new HO-1 inhibitors from three databases: Marine natural products (MNPs), ZINC natural products (ZNPs), and super natural II (SN2), providing an in silico proposal of imidazole-like compounds [75]. Many of these molecules have already been approved by the FDA or are known for possessing other activities, such as oceanapamine and verongamine, which have antibacterial and histamine H3-antagonist drugs, respectively [75].

### 3.3. Synthetic Small Inhibitory Molecules

HO expression is not limited to mammals, and several bacteria express a putative heme oxygenase enzyme (HemO), principally Gram-negative bacteria because the iron acquisition is critical for their survival and virulence [63,64]. HemO catalyzes the same reaction as human HO, although these two enzymes display structural differences [48,60]. Thus, *N. meningitidis* and *Pseudomonas aeruginosa* HemO show less than 15% homology as compared to human HO [48]. In fact, the active sites of HemOs from these two bacteria show a smaller solvent-accessible surface (7.5 Å) than human HOs (43.6–59.7 Å). Such differences suggest novel small antimicrobial molecules with bacteria-specific inhibitor potential over the human HO [48,60].

Furthermore, an alternative binding site has been identified in *P. aeruginosa* HemO [60,61,62]. A small molecule with a lipophilic group can allosterically inhibit the distal hydrophobic pocket responsible for catalyzing the initial hydroxylation of the heme group, impairing the stability and activity of the enzyme [66]. *Acinetobacter baumannii* [63], *Leptospira interrogans* [64], two Gram-negative bacteria, and *Clostridium perfringens* and *Corynebacterium diphtheriae* [65], two Gram-positive bacteria, also express HemO. This bacterial enzyme can also be targeted by these small molecules, inhibiting either the binding pocket to heme group or the alternative binding site [42,67]. A series of small molecules based on computer-aided drug design is shown in Figure 2C, with have proven capacity to inhibit HemO [48,66].

### 3.4. Inhibition by Genetic Engineering Approaches

Recombinant DNA technology development in the 1970s has allowed the generation of multiple tools for protein knockdown and gene knockouts, such as interference RNA (RNAi) and CRISPR-Cas9 [76]. Small interfering RNA (siRNA) and short hairpin RNA (shRNA) are two approaches for gene silencing using RNAi [77]. While the first leads to degradation of the sequence-specific mRNA target, the second one contains a loop structure that passes to siRNA processes, leading to the same result of degradation of the target mRNA [78].

On the other hand, clustered regularly interspaced short palindromic repeat (CRISPR) – Cas9 (CRISPR-associated nuclease 9) is a system adopted from bacterial defense mechanisms against viral infection [79]. Cas9 is complexed with a synthetic guide RNA, which is complementary to a specific nucleotide sequence of the DNA [80]. The target DNA sequence is recognized by the Cas9-RNA complex and induces double-strand breaks in the DNA with a subsequent repair, removal of genes, or addition of a new nucleotide sequence [80].

Both technologies are efficient in vitro, inhibiting HO-1 [49]. Even though no studies have used this technology for HO-2 inhibition, there is no doubt about its efficiency, given its high specificity and performance [81]. In vivo application of HO-1 siRNA and shRNA was also shown to knock down the enzyme, diminishing the growth of tumors [82]. Although no studies have evaluated in vivo CRISPR-Cas9 for knocking out HO-1, commercial kits for HO-1 knockout are available [49].

## 4. Therapeutic Implications of HO Inhibitors

The HO induction on immune cells has been extensively studied and reported effects over both innate and adaptive immunity [2,26]. The immunomodulation is associated with the increment of inhibitory products and the consumption of the pro-inflammatory heme group [4,5]. Thus, HO-1 induction in macrophages promotes the alternative activation towards an anti-inflammatory macrophage profile [83,84]. Besides, it has been shown that HO-1 modulates the production of IFN-β by macrophages and dendritic cells (DCs) [85] and inhibits the LPS-induced production of pro-inflammatory cytokines and inducible nitric oxide synthase (iNOS) [83,86]. On the other hand, HO-1 induction in DCs promotes a tolerogenic phenotype that contributes to the expansion regulatory T cells [87]. Along these lines, regulatory T cells constitutively express HO-1, and inhibition of this enzyme with ZnPP reduces their suppressor function in vivo [87]. Furthermore, in knockout mice, the absence of HO-1 correlated with a Th1 shift in cytokine responses and a predominantly pro-inflammatory state [88]. Moreover, HO-1 induction is also associated with a successful allogenic hematopoietic cell transplantation for acute leukemia [22,89].

However, high HO levels have been associated with elevated bilirubin loads [90], and HO inhibitors have been traditionally used as a treatment for ailments, such as neonatal jaundice [35]. Jaundice is an imbalance between the production and clearance of bilirubin, which can affect >80% of healthy newborns during the first week after birth [31,32]. Severe hyperbilirubinemia may occur and lead to bilirubin neurotoxicity if hyperbilirubinemia is unmonitored or untreated [31,32]. The efficacy of SnMP has been documented in clinical trials by contributing to decrease the duration of phototherapy [35].

Moreover, high levels of HO-1 have been detected in many pathologies and are associated with a deteriorated inflammatory response, leading to a decrease of the substrate heme and an increase of the anti-inflammatory products, which could impair the development of protective microbial immunity [44,45,46,47], cancer [39,40], or contribute to neurodegenerative diseases [41,42,43]. Along these lines, the administration of HO inhibitors can reduce these anti-inflammatory functions, promoting the development of an appropriate immune response [91,92,93,94]. Next, those pathologies resulting from excessive inflammation or an altered function of the immune system will be discussed.

### 4.1. Cancer

It has been reported that HO-1 is frequently overexpressed in various types of cancers, including adenocarcinoma, lymphosarcoma, and leukemia [8,95,96]. Furthermore, while the induction of this enzyme can protect from apoptosis in several cell types, its inhibition increases the susceptibility to oxidative stress [97]. In addition, HO-1 has been involved in regulating the cell cycle progression in a cell type-dependent way [98,99]. On the other hand, HO-1 induction can promote angiogenesis, which is fundamental during tumor growth and metastasis [100]. Accordingly, it has been suggested that higher levels of this enzyme can be associated with tumor growth and angiogenesis [8]. Furthermore, some cancer therapies can induce the expression of HO-1 in tumors [98,101].

Induction of HO-1 has effects on tumor cells and the tumor microenvironment, which include tumor-associated macrophages and other infiltrating cells. Furthermore, HO-1 expression contributes to the immune suppressive function of stromal macrophages [91,92]. It has also been observed that CD8^+^ regulatory T cells are HO-1^+^ and display an immunosuppressive activity in the peripheral blood and tumor of cancer patients [102]. These regulatory T cells expressing HO-1 have been involved in suppressing the immune responses against tumor cells [102]. In addition, M2 macrophages infiltrating malignant tissue express HO-1 [103] and display an anti-inflammatory profile critical for immune suppression [49]. Hence, in cases in which anticancer therapy is not effective due to the overexpression of HO-1, molecules that inhibit the activity of this enzyme might contribute to potentiate antitumor immunity [101,104]. As it was mentioned above, the expression of HO was shown to increase after chemo- and radiotherapy, which could contribute to decreasing treatment effectiveness [105]. Accordingly, the aggressive phenotype of rhabdomyosarcoma was shown to be reduced by treatment with HO-1 inhibitors [106], and other studies suggested that the use of ZnPP can increase the sensitivity and susceptibility to chemotherapy of hepatoma cells [39] and ex vivo samples obtained from multiple myeloma patients [107]. Moreover, ZnPP treatment reduces tumor growth and liver metastasis in a mouse model for neuroblastoma [108].

An anticancer effect was shown for an imidazole-based HO-1 inhibitor (SLV-11199) using human pancreatic and prostate cancer cell lines [40]. Treatment with SLV-11199 decreased the viability and migration of cancer cells and downregulated IL-8, MMP-1, and MMP-9, affecting also the epithelial to mesenchymal transition signaling axis [40]. In two thyroid cancer cell lines, both ZnPP and ketoconazole steadily arrested cellular migration and invasion [51]. These HO-1 inhibitors also effectively suppressed tumor growth in a xenograft mouse model, suggesting a therapeutic potential for these compounds [51]. In addition, the cytotoxic effect of imidazole and triazole derivatives was evaluated in prostate and breast cancer cell lines, observing selective effects over HO-1 or HO-2, consistent with an anti-proliferative capacity for HO inhibitors [109].

Furthermore, HO expression is related to the immunosuppressive function of dendritic cells and dysfunction of T cells [110,111]; thus, HO inhibition by shRNA or ZnPP improves dendritic cells maturation and revokes the unresponsiveness of CD4 and CD8 T-effector cells in neuroblastoma, glioma cells, and multiple myeloma [107,108,110].

### 4.2. Alzheimer’s Disease

Alzheimer’s disease (AD) is one of the most prevalent neurodegenerative disease characterized by extracellular amyloid plaques and intracellular neurofibrillary tangles in multiple brain regions of patients [112]. The innate immune cells inherent to the central nervous system (CNS) are microglia, equivalent to resident macrophages responsible for the defense against pathogens and cellular damage [113]. Hence, in AD and other neurodegenerative diseases, microglia-mediated neuroinflammation is considered as a pathology hallmark [114].

Such a pro-inflammatory and pro-oxidant environment promotes HO-1 expression, particularly in glial cells [41,115]. Consequently, the brain expression pattern of HO-1 has been determined in a murine model of AD [41]. HO-1 was preferentially located within the microglial compartment compared to other brain cells, and most of the microglial cells that overexpressed HO-1 were located near beta-amyloid peptide deposition [41]. Furthermore, HO-1 expression in microglial cells increased with aging and was even higher in a mouse model of AD [41]. It is thought that HO-1 is positively regulated, during physiological aging, in specific cells and areas of the brain that may be susceptible to stress. As a defense mechanism, based on the notion that CO can display an anti-apoptotic effect HO-1 expression has been proposed as a therapeutic strategy [116]. Nevertheless, the overexpression of HO-1 in a murine-like AD model causes cognitive decline and worsens disease progression [93]. Accordingly, a selective HO-1 inhibitor was studied in a widely accepted animal model of AD (APP_swe_/PS1_∆E9_ double-transgenic mouse), inducing amelioration of symptoms after treatment [117]. The overexpression of HO-1 was also detected in neurons and astrocytes of the cerebral cortex and hippocampus during AD [42,118], suggesting that the affected tissues are experiencing oxidative stress. The excessive production of CO may contribute to AD pathogenesis and to chronic oxidative stress [42,118]. Moreover, ferrous iron could produce oxidation of lipids, proteins, and nucleic acid in astrocyte mitochondria [119]. Furthermore, an HO-1-dependent Fe accumulation in microglia has been reported to produce neuroinflammation in aged mice [120]. Importantly, the ZnPP administration or the use of HO-1 knockout mice prevented the increase of inflammatory markers in the same model [120]. On the other hand, HO-1 overexpression in astrocytes induces a Parkinson´s disease-like phenotype [121]. All these observations suggest that HO-1 inhibitors can work as a potent therapeutic target for AD.

### 4.3. Infections

As described above, HO-1 plays a critical role at protecting the body from oxidative damage, which is supported by studies carried out in several inflammatory models [11,122,123]. Accordingly, HO-1-deficient mice are more susceptible to oxidant-induced tissue injury by endotoxin exposure [124]. Nevertheless, the involvement of HO-1 during microbial infections is less understood [125]. As mentioned above, the induction of HO-1 generally suppresses the inflammatory response and the production of anti-inflammatory cytokines, affecting both innate and adaptive immune responses. Therefore, although it is likely that an increased expression of HO-1 could impair microbial immunity, further research is required to better understand this phenomenon. Below we will review recent evidence shedding light on the understanding of the role of HO-1 during infections caused by various pathogens (Figure 3).

#### 4.3.1. Viral Infections

There is extensive evidence about the function of HO-1 during viral infections, and the induction of this enzyme has been associated with significant antiviral activity in a wide variety of infections [125,126,127,128,129,130,131]. Although several HO-1 effects have been described in some of them, evidencing direct mechanisms on virus components or mediating cellular responses, much remains to be elucidated in this regard [126]. The observation that myeloid cell conditional HO-1 knockout mice are prone to viral infections underscores the important role that this enzyme could play during the immune response against those pathogens [85]. Accordingly, it was reported that HO-1 induction reduces human respiratory syncytial virus (hRSV) replication in mice and lung inflammation, and these effects are at least partially associated with HO-1 induction in myeloid cells [129]. In fact, treatment with an HO-1 inhibitor (SnPP) increased hRSV mRNA copies post-infection [129]. On the other hand, HO-1 induction with CoPP decreases the expression of herpes simplex virus (HSV) virus-proteins in epithelial cells by affecting events downstream of the virus attachment to the cell surface [130]. Accordingly, HO-1 inhibition might be detrimental to virus clearance. Hence, the administration of HO-1 inhibitors can counteract the reduced replication associated with HO-1 inductors [131,132].

#### 4.3.2. Bacterial Infections

The contribution of HO-1 during bacterial infection could be beneficial or harmful depending on the infecting pathogen, as pointed out below in the section. Myeloid cells are one of the first lines of defense, and they constitute critical players during an immune response to bacterial infections [133]. Accordingly, the performance of these cells can profoundly impact the efficiency of adaptive immunity. Among them, the phenotype of macrophages contributes to the development of a proper immune response against bacteria. Therefore, HO-1 overexpression in macrophages has been associated with alternative activation characterized by an anti-inflammatory phenotype known as M2 [103]. Thus, *Helicobacter pylori* infection leads to an increased HO-1 expression in macrophages and a mixed M1 classical/Mreg profile, which favors bacterial survival [46]. However, HO-1 inhibition with CrMP produces an increased polarization towards the M1 phenotype and a reduced Mreg phenotype [46]. Similarly, the genetic ablation of HO-1 in infected mice produces an enhanced M1/Th1/Th17 response and reduced *H. pylori* colonization [46]. In a similar manner, *Burkholderia pseudomallei* infection also induces HO-1 expression in macrophages, impairing intracellular clearance by increasing TNF-α, IL-6, and MCP-1, and reducing IFN-γ secretion [94].

On the other hand, induction of HO-1 expression has been reported after *M. tuberculosis* infection, which was reduced after antibiotic administration [134,135]. However, HO-1 knockout mice are more susceptible to *M. tuberculosis* infection [136]. Furthermore, administration of a well-characterized HO-1 inhibitor (SnPP) to mice during an acute *M. tuberculosis* infection produced a significant reduction in pulmonary bacterial loads compared to conventional antibiotic therapy [137]. Additionally, it has been recently reported that the inhibition of HO-1 with SnPP induces an IFNγ response, NOS2 expression, and NO production during *M. tuberculosis* infection [44]. Importantly, the effect of SnPP was Fe^2+^ dependent, and the Fe administration suppressed the T cell-dependent anti-bacterial effect [44]. Similarly, HO-1 induction impairs the elimination of the *Mycobacterium abscessus*, and inhibition of this enzyme increases phagosome-lysosome fusion in macrophages [138]. On the other hand, HO-1 inhibition in macrophages during *Salmonella* infection stimulates the anti-bacterial immune effector pathways and promotes bacteria elimination [139].

The contribution of HO inhibitors has also been evaluated in a murine model of pneumonia-induced sepsis [140]. Neutrophil migration to the bronchoalveolar space is increased by treatment with a nonspecific HO inhibitor (ZnPP), improving the clearance of *Klebsiella pneumoniae* and causing a decrease in the systemic inflammatory response and alveolar collapse [140]. Importantly, as mentioned above, HO-1 modulation has effects on the innate and adaptive response. However, most of the studies have focused mainly on cells of myeloid lineage. Thus, the beneficial impact of using HO inhibitors during infections could also be associated with other elements of the adaptive response.

#### 4.3.3. Parasite Infections

Parasite infections turn out to be a major human health problem in third world countries [141]. Two types of response to infections with parasites have been described, resistance and tolerance [142]. The chronicity of the disease with diverse parasites suggests the development of an immunosuppressive process during the infection [143]. Notably, it was shown that the activity of HO-1 could interfere with the resistance process, both by the elimination of heme and by the products of this enzymatic reaction [144]. Thus, the induction of HO-1, on the one hand, impairs phagocytosis and, on the other, reduces the secretion of pro-inflammatory cytokines [143]. Recently, *Trypanosoma brucei* has been reported to suppress the host pro-inflammatory response through the secretion of specific aromatic ketoacids that activate the Nrf2/HO-1 pathway in macrophages [143].

Interestingly, several parasitic infections are supported by the establishment of tolerance, thus in various models, HO-1 induction reduces the severity of the disease but does not reduce the parasite load in malaria [145]. Therefore, it is likely that the CO produced by HO-1 in macrophages can prevent the disruption of the brain-blood barrier [146]. On the contrary, malaria severity positively correlates with the HO-1 levels in white blood cells [147], triggering immunosuppressive functions on antigen-presenting cells [145]. Furthermore, cobalt protoporphyrin (CoPP), an HO-1 inductor, increases *Leishmania infantum chagasi* burden in human and mouse macrophages, and HO-1 was associated with visceral leishmaniasis (VL) disease susceptibility as the VL patients presented higher systemic concentrations of HO-1 than healthy individuals [148].

On the other hand, the infection with *Fasciola hepatica* helminth has been shown to induce HO-1 expression in mice and pharmacological induction of this enzyme with CoPP increases clinical signs associated with the disease [45]. Besides, the *F. hepatica* infection promotes HO-1 expression in DCs and macrophages, promoting on these cells a tolerogenic profile and an alternative activation phenotype, respectively [45]. Thus, the upregulation of HO-1 leads to an increase of IL-10 and TGF-β, promoting and benefiting the arrival of parasites to the liver [45]. Conversely, HO-1 inhibition with SnPP protects mice from the infection and significantly decreases the levels of IL-10 and TGF-β [45].

#### 4.3.4. Fungal Infections

Various fungal organisms are part of the gastrointestinal flora of healthy people, but they have the potential to become invasive pathogens. For instance, in patients with impaired immune response, *Candida albicans* can colonize specific host organs and establish disseminated bloodborne disease (candidemia) with high morbidity and mortality [149]. Hemoglobin induces the *C. albicans* HO activity, and it was hypothesized that HO expression was able to induce *C. albicans* growth in mammalian hosts [150]. Furthermore, in a mouse model of disseminated candidiasis, *C. albicans* HO activity and CO production limit the host immune response and contribute to the survival and virulence of *C. albicans* [47].

## 5. Discussion

Several studies have focused their attention on the anti-inflammatory and antioxidant properties of the products generated by the HO system, especially HO-1 since it is the inducible isoform [11,12,151]. However, as described in this revision, HO overexpression can also be detrimental to many pathologies. Here, we reviewed an immunomodulatory perspective of the HO inhibition.

Oxidant and pro-inflammatory environments enhance the transcription of HO-1, and an increase of heme improves HO-2 activity [11,152]. In cancer, the immune-inflammatory cells infiltrated in tumor cells can modulate the tumor progression, actively participating in nutrition, angiogenesis, invasion by matrix metalloproteinases secretion, and decrease the antitumor response [153]. Several studies have addressed the overexpression of HO-1 in macrophages infiltrating tissues and how HO-1 inhibition by first- and second-generation inhibitors can reduce the aggressive phenotype in certain types of tumors [8,13]. Although studies center their attention on HO-1 inhibition, the inhibitors used are non-selective for all the isoforms, and the implication of HO-2 in cancer remains unknown.

On the other hand, neuroinflammation is accompanied by overexpression of HO-1 [154]. Consistent with this notion, activated microglia are present during AD, overexpressing the enzyme and co-localizing with the sites of beta-amyloid peptide deposition [155]. In Parkinson’s disease, another neurodegenerative pathology, overexpression of HO-1 was shown in astrocytes near the substantia nigra [43]. Thus, HO-1 has been suggested as a potential therapeutic target for drug development for neurodegenerative diseases. Besides, this enzyme can be considered as a biomarker for AD and other neurodegenerative diseases. Additionally, as HO-2 expression is constitutively high in the brain, this isoform can likely play an important role in neuroinflammation, and selective inhibition may be a potential treatment to be considered in future studies.

In the context of infectious diseases, the impact of HO-1 inhibition remains not fully understood. In viral infections, the induction of HO-1 reduces hRSV replication [126,129] and decreases the expression of HSV proteins [130]. Accordingly, HO inhibition in viral infection can be detrimental, but it promotes a favorable immune response in many bacterial infection responses [44,94]. Furthermore, in several parasitic and fungal infections, the inhibition of HO-1 blocks the induction of tolerance, promoting pathogen clearance [47,145]. Interestingly, inhibition of bacterial HemO has also been explored as an antimicrobial treatment to approach the current antibiotic resistance problem. Importantly, the inhibition of HemO is pathogen specific with no significant effect on the host HO activity [48,63,64].

The medical applications discussed in this article focused on the inhibition of the HO-1 isoform. However, two details are systematically skipped: (1) most of the inhibitors used are non-selective between isoforms, and (2) HO-2 activity is affected by posttranslational modifications (as is the isoform with HRM cysteine-residues regions). Nitrosylation reaction can inhibit up to 60% of the enzyme activity, reached by sodium nitroprusside (SNP) treatment [21]; in contrast, SNP induces the expression of HO-1 [156].

Another important point to consider is the potential adverse effects produced by the administration of an inhibitor treatment. Complications using PPs and MPs, such as oxidation of structures and metabolites (membrane lipids, energy metabolites, proteins, and nucleic acids), as well as the hepatotoxicity caused by the imidazole ring in second-generation inhibitors, have been described [36,38,73,109]. Forthcoming HO inhibitors must consider this point for future studies and potential clinical applications.

Most of these new approaches explore selective inhibition of HO-1 and HO-2 isoforms, and many compounds from the second-generation inhibitors show a better affinity for one over the other isoform. Still, genetic engineering, such as RNAi and CRISPR-Cas9, allow isoform selectivity, tissue-specific inhibition, and in the CRISPR-Cas9 case, a total knockout of genes, representing an exceptional tool for future inquiries in the HO system inhibition [157].

## Figures and Tables

**Figure 1 ijms-22-00294-f001:**
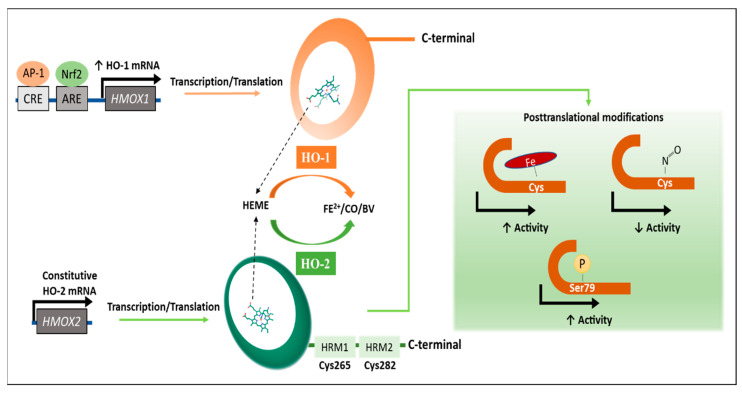
Regulatory mechanisms for the expression and activity of HO enzymes. Heme oxygenase 1 (HO-1) is induced by transcription factors, such as Nrf2 and AP-1, as schematically shown in the figure. Heme oxygenase 2 (HO-2) binds to Fe^2+^ ion of the heme group through cysteine residues in the heme regulation motifs (HRMs) of the enzyme, inducing its activity (up-arrow). Additionally, phosphorylation of serine 79 residue enhances the enzymatic activity (up-arrow). In addition, nitrosylation of the cysteine residue inhibits enzyme activity (down-arrow). Both isoforms catalyze the oxidation of the heme group producing Fe^2+^, CO, and biliverdin (BV).

**Figure 2 ijms-22-00294-f002:**
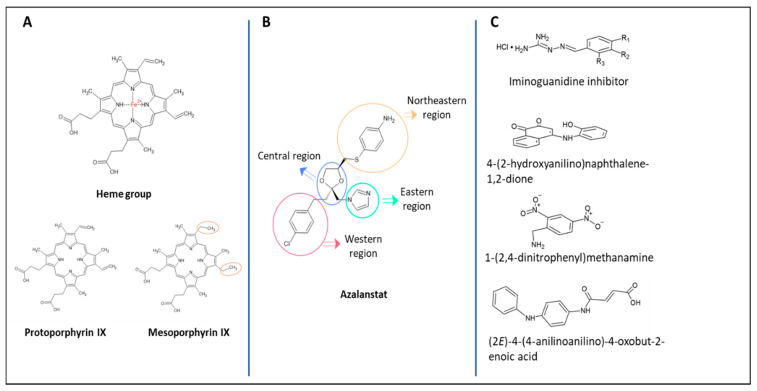
HO system inhibitors. (**A**) The first generation of HO competitive inhibitors are based on the protoporphyrin structure shared by the heme group. (**B**) Azalanstat structure, basis of the second generation of non-competitive HO inhibitors. Central and eastern regions originate the imidazole-dioxolane compounds. (**C**) Small molecules that inhibit bacterial heme oxygenase (HemO), differences in the heme-binding pocket between the pathogen and host allow these approaches to act as bacteria-specific antimicrobial drugs.

**Figure 3 ijms-22-00294-f003:**
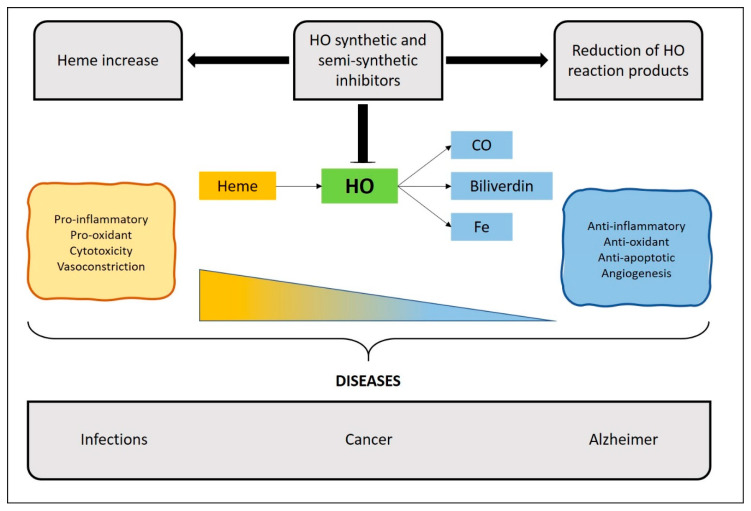
Schematic representation of the effect of HO enzyme inhibition in immune-mediated diseases.

**Table 1 ijms-22-00294-t001:** Inhibitors of the HO system. General description of first-, second-generation, and novel HO inhibitors.

HO System Inhibitors	Name	Inhibitors Characteristics	References
First generation inhibitors	Metallo-protoporphyrins: Sn-, Zn-, Mn-	Competitive inhibitors Non-selective isoforms Present photo-reactiveness	[23,33,38,51,52,53]
	Metallo- mesoporphyrins: Cr- and Mn-	Less or non-photo-reactiveness	
Second generation inhibitors	Azalanstat-derived imidazole-dioxolane compounds	Non-competitive inhibitors Non-selective isoforms	[38,54,55]
	Imidazole-derived antifungal agents: ketoconazole, terconazole, and sulconazole	More selectiveness against HO-1 isoform.	[51,56,57]
	Clemizole and derive compounds	First HO-2 isoform selective inhibitor	[58,59]
Small molecules inhibiting microbial HO	Small antimicrobial molecules against heme oxygenase (HemO) expressed by microbes.	Selective inhibition of putative HemO express by microbes.	[48,60,61,62,63,64,65,66]

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
