# Peer review of "Immune Modulation by Inhibitors of the HO System"

_ijms, 2020, doi:10.3390/ijms22010294_

Round 1
Reviewer 1 Report
In the manuscript entitled “Immune modulation by HO system inhibitors” Fernandez-Fierro and coworkers discuss how the inhibition of selective HO isoforms exert beneficial effects in immune-mediated disease analyzing this aspect in cancer, Alzheimer disease and infections. This topic is very interesting, but the manuscript does not investigate in deep on the immunomodulatory aspects of the inhibitors of HO system so the manuscript should be revised and reorganized highlighting these aspects.The connection between diseases discuss in the manuscript, heme oxygenase and the involvement of the immune system in relation to the role of HO inhibitors is not well addressed.
The paragraph relating to inhibitors is complete; on the contrary the paragraph 4, should be reorganized focusing the attention on the role of HO inhibitors on immune cells involved in the considered disease (cancer, Alzheimer o infections).
In paragraph 4.1 (Cancer section) the role of HO-1 inhibition (ref. 84 and 85) is referred to the enhancement of anticancer treatments such as photodynamic and chemo-therapy and not to the effect on immune cells.
Furthermore, the role of HO-1 inhibitors on tumor cell proliferation, migration and invasion was discussed, but no mention of HO-1 involvement in tumor immuno-escape.
These aspects should be revised in all section entitled “Therapeutic implications of HO inhibitors in immune-mediated disease”. Moreover, the title of this section should be revised.
Table regarding HO inhibitors should be added.
Minor comments:
The part relating to the action of the metabolic products of HO-1 activity must be moved from paragraph 2 to paragraph 1 (line 46).
The paragraph relating to the regulation of HO activity must be dealt with in more detail or combined with paragraph 1. In the same paragraph line 75 “HO-2 has” should be write in correct style
Paragraph 3.1 line 110 must be correct since Cobalt protoporphyrin is a strong inducer oh HO-1 and not an inhibitor (Willix JL et al Activation of Heme Oxygenase Expression by Cobalt Protoporphyrin Treatment Prevents Pneumonic Plague Caused by Inhalation of Yersinia pestis - Antimicrobial agents and chemotherapy 2020; Katori M et al Cobalt protoporphyrin-induced overexpression of heme oxygenase-1 protects rat hearts from cold ischemia/reperfusion injury via anti-apoptotic pathway – JACS 2000)
English should be revised
Reviewer 2 Report
This is a well written and very interesting overview of the potential impact of heme oxygenase inhibition (HO-1 and/or HO-2) in several diseases in which HO overexpression may be deleterious.
The manuscript is clear enough and embellished with figures. I have only minor comments:
About regulation of the HO activity (paragraph 2): a brief but more exhaustive description of the regulation of HO-1 transcription should be added including the roles of STAT3/IL-10/MAPK and NF-kB. Polymorphisms (SNPs) of HO-1 gene are briefly mentioned but nothing about the (GT)n repeat length polymorphism in the promoter region of the HO-1. Importantly, a long (GT)n repeat has been correlated with a lower transcription of HO-1 and higher inflammation and worse prognoses of several diseases. On the opposite, lower transcriptors of HO-1 have a stronger graft versus leukemia effect and have less disease relapse after allogeneic hepatopietic stem cell transplantation (ref: Kollgaard T, Kornblit B, Petersen et al. (GT)n Repeat Polymorphism in Heme Oxygenase-1 (HO-1) Correlates with Clinical Outcome after Myeloablative or Nonmyeloablative Allogeneic Hematopoietic Cell Transplantation. PLoS One. 2016;11(12):e0168210 or (perhaps less convincing) Yinghao Lu et al. Identification of heme oxygenase-1 as a novel predictor of hematopoietic stem cell transplantation outcomes in acute leukemia. Cell Physiol Biochem 2016;39:1495-1502. A recent relevant publication in the JCI insight showing the benefit to inhibit myeloid HO-1 in anti-tumor vaccination protocol might be added (JCI Insight: 2020 Jun 4;5(11):e133929. doi: 10.1172/jci.insight.133929. Heme oxygenase-1 orchestrates the immunosuppressive program of tumor-associated macrophages.
About first generation of HO-1 inhibitors (paragraph 3.1) : authors describe the lack of selectivity but the given reference is related to HO-2 only. Moreover, HO-1 expression has been shown to be paradoxically induced by some HO inhibitors of the first generation. This could be mentioned.
About cancer (paragraph 4.1, line 224) : References deal with in vitro tumor cell lines and it seems there is no reference with in vivo enhanced antitumor immunity weakening the conclusion.
HO-1 in Alzheimer's disease is very complex. The authors state that HO-1 could lead to oxidative damages in Alzheimer's disease and that HO-1 inhibition attenuates oxidative damages and alzheimer's disease model in vivo. However, only one reference in the paper is mentioned and this should be strengthened by other references.
In the discussion: What is the meaning of MPP ?
Round 2
Reviewer 1 Report
The authors improved the manuscript as request